# Are Services Inclusive? A Review of the Experiences of Older GSD Women in Accessing Health, Social and Aged Care Services

**DOI:** 10.3390/ijerph17113861

**Published:** 2020-05-29

**Authors:** Tinashe Dune, Jacqueline Ullman, Tania Ferfolja, Jack Thepsourinthone, Shirali Garga, Zelalem Mengesha

**Affiliations:** 1Translational Health Research Institute, School of Medicine, Western Sydney University, Penrith 2751, NSW, Australia; T.Dune@westernsydney.edu.au (T.D.); 18718847@student.westernsydney.edu.au (S.G.); 2School of Science and Health, Western Sydney University, Penrith 2751, NSW, Australia; J.Thepsourinthone@westernsydney.edu.au; 3Centre for Educational Research, School of Education, Western Sydney University, Penrith 2751, NSW, Australia; J.Ullman@westernsydney.edu.au (J.U.); t.ferfolja@westernsydney.edu.au (T.F.)

**Keywords:** LGBTIQ, gender and sexually diverse, women, health care, aged care, social care

## Abstract

The review aimed to examine the views and experiences of ageing gender and sexually diverse (GSD) women—a triple minority in relation to their age, gender and sexual orientation—in accessing health, social and aged care services. Eighteen peer reviewed articles identified from seven electronic databases in health and social sciences were evaluated according to predefined criteria and a thematic review methodology drawing upon socio-ecological theory was used to analyse and interpret the findings. Four major themes were identified from the analysis: “The Dilemma of Disclosure”, “Belonging/Connection”, “Inclusiveness of Aged Care” and “Other Barriers to Access Care”. In the dilemma of disclosure, older GSD women consider factors such as previous experiences, relationship with the provider and anticipated duration of stay with the provider before disclosing their sexual identifies. The review also revealed that aged care services lack inclusiveness in their policies, advertising materials, aged care spaces and provider knowledge and attitude to provide sensitive and appropriate care to GSD women. Overall, older GSD women experience multiple and multilevel challenges when accessing health, aged and social services and interventions are needed at all levels of the socio-ecological arena to improve their access and quality of care.

## 1. Background

The term ‘older age’ in the context of this paper refers to individuals aged 55 years and over [1]. Globally, the ageing population accounts for 8.3% (608 million) of the total population and has increased by 16% (84 million) since 2010 [2]. Additionally, it is expected that the ageing population will have more than doubled by the year 2050 [3]. In countries such as Australia, China, the UK, and the USA, the life expectancies for men and women are estimated to be over 75 and 80 years, respectively [2]. Among this growing minority, however, another minority exists—ageing LGBTQI (lesbian, gay, bisexual, transgender, queer and intersex) individuals. In Australia for example, 27.3% of gay men and 19.4% of lesbian women are amongst the ageing population [4].

Described as “an invisible minority”, the concerns and wellbeing of ageing LGBTQI individuals can often be overlooked [5]. Ageing individuals are often perceived as desexualised—asexual—and the effects of homophobia are compounded when an individual’s LGBTQI identity is revealed [6]. Discrimination (both actual and expected) is identified as one of many sources of concern for ageing LGBTQI individuals [7]. Further still, when asked about their attitudes towards engaging with health and social service providers as they age, a higher proportion of lesbian women in comparison to gay men expressed doubt in their potential access to non-discriminatory treatment [7]. It has been identified that actual and/or expected discrimination are one of three major barriers hindering LGBTQI individuals’ disclosure of information when dealing with health care services [8]. Additionally, a study by Kahn and colleagues [9] revealed that over half of the health care professionals surveyed were uncomfortable dealing with issues relating to gay and lesbian individuals (e.g., sexually transmitted infections). This presents a valid reason to assess the needs and experiences of ageing LGBTQI individuals in accessing health, social, aged care and retirement services.

Similarly, it has been argued that both staff and residents in aged care facilities are potential sources of sexuality-related discrimination [10]. A study by Hinrichs and Vacha-Haase [6] found that aged care staff members tended to rate romantic relationships relating to same-sex couples more negatively than opposite-sex couples. Another study interviewed heterosexual aged care residents on their reactions towards another resident’s hypothetical coming out [11]. Their findings indicated that most residents would behave negatively towards same-sex attracted residents in the form of maintaining distance from the resident and extreme rejection. Along with Hinrichs and Vacha-Haase’s [6] findings, this further supports the assertion that both staff and residents are sources of discrimination [10]. It is no surprise that ageing LGBTQI individuals often express a desire for either LGBTQI-exclusive or LGBTQI-friendly aged care services [10].

Relative to this, a study by Horner et al. [12] examined aged care and retirement services in relation to their inclusion/address of LGBTQI issues. It was revealed that few accommodated the specific needs of LGBTQI individuals. Additionally, in a recent interview study by Barrett and colleagues [13], participants reported that the lack of LGBTQI-inclusive services forces them to enter a heteronormative context and go back into the ‘closet’. Given these concerning findings regarding LGBTQI-inclusive policies and practices in regards to health, social, aged care, and retirement services [8,10], it can be argued that a synthesised examination of all these services is required in order to obtain an up-to-date overview of these sectors and provide recommendations to address the issues presented.

The above stated issues are compounded for ageing LGBTQI women. Research indicates that LGBTQI women’s health and wellbeing outcomes are poorer than that of their male and heterosexual counterparts and that these outcomes are even worse for older women. A 2009 survey of 371 gay and lesbian-identifying aged care residents found that over 45% were concerned about experiencing sexuality-based discrimination when entering aged-care. Of these participants, significantly more women (75%) harboured concerns about not finding LGBTQI-specific residential care later in life [14]. This is partially the consequence of poorly informed and ill-equipped health and community services resulting in discrimination or exclusion of LGBTQI women. Accordingly, it is not surprising that LGBTQI women are less open about their sexuality and more anxious about accessing services as a result of heteronormative assumptions made by health and community service professionals which render LGBTQI older people “invisible” [15]. Case in point: a recent survey of aged-care residential providers found that nearly 90% of participants were unaware of having ever served LGBTQI clients [16] and fewer than half of a stratified random sample of 409 general practitioners in an Australian study reported feeling comfortable providing care for gay or lesbian patients [9]. These limitations point to the breadth and gendered nature of the issue within society. This in turn causes GSD women to avoid health services and increases social isolation.

Currently, there exists a number of reviews that focus on the health, social, aged careand/or retirement service experiences of LGBTQI individuals [17,18,19]. However, no synthesised report exists on the access of health, social, aged care, and retirement services in relation to ageing LBTQI women, specifically a “triple minority” [20] in relation to their age, gender, and sexual orientation. The current review aims to provide a concise overview of the current literature detailing the experiences/perspectives of ageing LBTQI women in accessing health, social, aged care, and retirement services. Findings of the review may contribute to engaging and informing decision makers to redesign health, social and aged care services to make them more accessible to older LBTIQ women and providing insights to foster appropriate care-seeking behaviour.

## 2. Methods

### 2.1. Research Design and Search Strategy

This is a systematic review aiming to collect, analyse and synthesise secondary data and provide a detailed summary of current evidence relevant to the research objective. Peer reviewed articles were identified from seven electronic databases in health and social sciences: Scopus, ProQuest central, PubMed, PsycINFO, Informit, ProQuest social science database and SocINDEX with full text. The search strategy was developed after defining the review question using PICo: Population, Interest and Context [21]. An initial search of articles revealed that literature on the health, social and aged care services experiences of older GSD women started to emerge around 1990. As such, the year of the search was agreed to start from this period. In this process keywords and index terms used to describe papers relevant to the aim of the review were identified and organised (Table 1). The search strategy was discussed, modified and approved by authors T.D., Z.M., J.U. and T.F. Following a preliminary search in ProQuest for key words in the title, abstract, and full text, relevant findings were limited amongst the results. Relevant key words were found in the titles with the abstract and full-text searches not meeting the inclusion criteria. Literature searches were then conducted across the identified databases using the key words within the title field. The results were downloaded and saved in an Endnote library for further screening and examination. Finally, the reference list of all identified reports and articles were examined for additional literature.

The following inclusion criteria were then applied to identify the most relevant articles for the review: primary research including qualitative, quantitative and mixed method designs; studies which examined the target population’s experiences (i.e., views and perceptions including facilitators and barriers) of accessing health, social and aged care services; papers published after 1990 (Table 1). Only a handful of studies initially addressed older gender and sexually diverse (GSD) (lesbian, transgender, transsexual, bisexual, intersex and queer) women’s views and experiences of accessing health, social and aged care services. As a result, the inclusion/exclusion criteria were modified to incorporate studies that included both older and younger gender and sexually diverse women. In total, 141 articles were assessed with 18 remaining in the final review after the exclusion criteria were applied [22]. In this review we used Atkinson et al.’s standards for reporting literature searches, indicating inclusion criteria and making research syntheses more transparent and easy to replicate [22].

### 2.2. Quality Assessment

The majority [17] of the studies included in this review involved qualitative approaches. However, there are wide variations in understandings and criteria of what constitutes quality in qualitative research, which makes quality assessment challenging [23,24]. In addition, previous authors reported the dearth of research focusing on the health care experiences of the GSD population in accessing much needed services such as health, social and aged care [25]. The secondary aim of this review was also to understand what research exists concerning the views and experiences of older GSD women in accessing health, social and aged care services. As such, quality assessment of the included studies was found to be less relevant despite the systematic search and identification of relevant literature.

### 2.3. Data synthesis and Reporting

Health, social and aged care provision requires the involvement and coordination of several micro systems in the broader health system [26]. As such, a systematic approach which considers influences beyond the individual level is relevant to have a broader understanding of the factors that impact care access and utilisation [26]. The socio-ecological theory is most central to this process as it recognises multiple domains of influence in an individual’s social environment that impact health, social and health care access and provision [27,28]. With this in mind, we used the socio-ecological model as a framework to systematically analyse experiences of LBTQI women in accessing health, social and aged care services at four levels: individual (micro), interpersonal (meso), organisational (macro), and societal (exo) levels.

A theoretical thematic analysis according to the approach described by Thomas and Harden [29] drawing on Bronfenbrenner’s socio-ecological framework was used to analyse relevant data obtained from the 18 included articles. Initially, the first author read and re-read the extracted data and developed first-order codes such as “Health conditions”, “Discrimination”, “Negative provider attitude” and “Acceptance in aged care”. An inductive approach to coding and analysing the data was used, and first-order codes were identified using the language that was being coded. At the same time the second author also read the extracted data. The coding frame was then discussed and amended in a joint meeting. This was followed by coding of the entire data set using Quirkos (Quirkos, Edinburgh, Scotland, UK), a qualitative research software program which facilities organisation and management of qualitative data. Codes were then collapsed into conceptual themes such as “Is aged care inclusive?”, “The dilemma of disclosing sexual identity”, “Connection/Belonging” and “Barriers to care” were identified through a process that involved examining similarities, differences and patterns across the coded data.

## 3. Results

From the 141 potentially relevant articles identified, 18 articles were included in the systematic review (Figure 1).

### 3.1. Basic Characteristic of the Included Studies

Table 2 shows the demographical and methodological characteristics of the included studies. These studies represent the experiences of 830 GSD women in accessing health, social and aged care services. Most studies were conducted in the USA (50%) and Canada (27.7%) with only 16.6% and 5% conducted in Australia and Norway respectively. Regarding the study population, fifteen (83%) studies involved only lesbian women and three others included both lesbian and bisexual women. Seventeen of the studies employed qualitative research designs with in-depth and semi-structured interviews being the most commonly reported methods of data collection. Most of the studies focused on the experiences of health (8), aged care (3) and home care (4). Various theoretical frameworks were used in the studies, with eleven of them involving one of the following: feminist poststructuralism (2), psychological contracting (1), ecological perspective (1), phenomenology (1), grounded theory (1), socio-linguistic (1), feminist political economy framework (1), heteronormativity theory (1), feminist ethic of care (1) and intersectionality (1).

Given that some studies included the experiences of GSD women as young as 18 years old, only the findings as they relate to older GSD women were extracted where possible. However, not all papers delineated between the results of older and younger participants making extraction of only those experiences from those papers difficult. The implications of this are further discussed in the limitations section of the paper.

Ultimately, three major themes and twelve sub-themes were identified after line-by-line coding of the extracted data from each of the studies included in the systematic review. The three major themes were: *The Dilemma of Disclosure, Is Aged Care Inclusive* and *Systemic Barriers to Care*. The below Figure 2 summarises the findings of the study according to the theoretical approach employed to present and interpret the findings.

### 3.2. Interpersonal Level: The Dilemma of Disclosure

This first major theme focuses on older GSD women’s experiences of disclosing their sexual identity with health care professionals, home care workers, employers, patients and colleagues at work. It is organised under two subthemes: negotiating disclosure and concealing sexual identity.

#### 3.2.1. Homophobia

Several articles reported that older lesbian and bisexual women experienced homophobic and heterosexist attitudes and responses by health and home care workers in the system when they tried to access services [30,34,46], and those who did not experience these attitudes realised that they had not disclosed their sexual and gender identities to the providers. These homophobic attitudes and reactions were reported to have several impacts on lesbian and bisexual women’s health and access to care. For instance, it was reported that older lesbian women remained closeted to their care providers due to fear of homophobic attitudes and reactions [46]. Further, older lesbians either abstained from or delayed using formal health services and discontinued home care due to health care providers’ homophobic and heterosexual attitudes and reactions. Homophobia and the need to monitor provider reactions to sexual identity was also seen to be a stressful and energy-sapping burden for lesbian women, particularly for those battling an underlying condition or disease [34,46].

#### 3.2.2. Negotiating Disclosure

Many of the older GSD women represented in these studies detailed the importance of disclosing their sexuality diversity to their care providers, describing this as “crucial to their sense of self” [38] and stressing that they took “no efforts to hide it” [34]. However, the decision to come out was a difficult process influenced by several factors. For example, Barbara and colleagues [32] and Grigorovich [34] stated that lesbians disclosed their sexual orientation when there is a positive relationship with their providers and they see welcoming messages and attitudes from them. In addition, Grigorovich [34] identified that lesbians would disclose their sexual identity when they expected to receive care from the same healthcare provider for a long period of time. Grigorovich [34] also added that previous experiences of prejudice and discrimination influenced lesbian and bisexual women’s decision-making with regard to disclosing sexual identity.

After disclosure, lesbian and bisexual women experienced both positive and negative reactions from the health care professionals and home care workers. For example, Bjorkman and Malterud [33] reported that lesbians were pleased as their lesbian identity was acknowledged by doctors and the information was considered important, which could influence their treatments. Other studies also reported that different categories of health professionals and home care workers were supportive and accepting of lesbian identities following disclosure [34,38]. On the other hand, a majority of the studies reported that lesbian and bisexual women experienced negative reactions from health care providers and home care workers following disclosure of their sexual orientation. These included prejudice [32,33,34], association of all illness with the lesbian or bisexual identity [33] and discrimination [34]. In some instances, health care providers discontinued clinical examination after learning their patient’s sexual identity, and another provider had to come to finish that examination [32]. Another study also reported that health care providers made homophobic remarks [38] that upset the women. These reactions were reported to have negative consequences on the women’s health and wellbeing. For example, Grigorovich [34] reported that providers’ negative reactions to the women’s sexual identity had a significant role in the women’s “feeling of isolation and being silenced”. In addition, some studies revealed that lesbian women changed their doctors after experiencing negative reactions [33].

#### 3.2.3. Concealing Sexual Identity

On the other hand, several of the studies included in the review reported that older lesbian and bi-sexual women chose not to disclose their sexual identity when interacting with health care professionals [32], home care workers [34], employers [30,32] and colleagues at work places [38]. Older lesbian and bisexual women’s hiding of their sexual identity from health care professionals and home care workers was due to fear of judgment and prejudice [32], heterosexual assumptions [34] and fear of rejection by health care professionals or lack of care [30]. Other studies reported that fear of negative consequences on their career developments [38], including the risk of losing their jobs [30], were the significant reasons not to disclose their sexual identities. For some older lesbian and bisexual women, sexuality was not a topic of discussion at their work places or considered by them to be a central part of their identity [38]; specifically, lesbian health care professionals believed that disclosing their sexuality identity was not relevant to their patients [38]. However, the decision not to come out was not without consequences. A number of studies reported that older lesbian and bisexual women who concealed their sexual identities from practitioners experienced reduced happiness and self-esteem [39]. In one study, consequences included women being made to pay for unnecessary, heteronormative medical investigations related to sexual and reproductive health such as pregnancy tests [32].

#### 3.2.4. Connection/Belonging and Support-Seeking Behaviour

This theme identified from the analysis concentrated on the feelings and experiences of older GSD women in regards to their connection and belonging with their family and friends, other GSD people and the wider community. The discussion also includes how these connection experiences impact the delivery and receipt of care for lesbian and bisexual women.

#### 3.2.5. Non-GSD Friends and Family

Several of the studies included in this review reported a mixed experience in relation to lesbian and bisexual women’s connection/belonging with their families and non-GSD friends and their experiences of social support. In their studies, for example, Averett and colleagues [31] and Richard and Brown [43] stated that lesbian women were out to their family members and non-GSD friends and had a positive relationship with them. In addition, these women had regular contact with their children and other family members [31], and consequently, the women either preferred or received care and support from their family members and friends when becoming sick or disabled [31,40,43]. On the other hand, Reference [35] reported that many lesbian and bisexual women experienced homophobic attitudes and behaviour from their family members, and as a result, they were not interested in consulting with them about challenges to their health or social support. Furthermore, lesbian participants in a study by Rowan and Giunta [44] discussed experiencing difficulties in their sexuality diversity being accepted within their family and social circles. As a result, they emotionally and physically distanced themselves from their family members [39]. In addition, lesbian and bisexual women’s friends were reported to have health conditions including disability, which made it difficult for these women to connect and seek support [35,43]. Those women who had partners managed to get care and support from their partners, but single women were left without care, except from the occasional support from neighbours and friends [35].

#### 3.2.6. GSD Community

A majority of the studies described that most of the time older lesbians made connections with other lesbians because they considered it as a ‘safe haven’ [30,42]. Averett, Yoon and Jenkins [31] also reported that lesbian women’s closest friends were other lesbians and heterosexual women within 10 years of their age. This connection helped them to form a strong informal support network which would look after them when needing care and support [37,43,45]. Lesbian women who were living in a community housing alongside other lesbian-identifying women formed a strong connection and support system that provided the necessary care women needed during illness. These women also found lesbian-friendly health professionals [37,40]—especially as age-related health concerns increase with age. Only a single study explicitly highlighted lesbian women’s connections to gay-identifying men as a source of social support [45]. Despite connection with other lesbians being a ‘safe haven’, some lesbian women experienced discrimination based on their age, as most of the lesbian-only programs and support systems were inclined towards fulfilling the needs and aspirations of younger lesbians [39], and younger lesbians were not interested in interacting and socialising with their older counterparts [39].

#### 3.2.7. General Community

Overall, the general social environment was reported to be hostile to older same-sex attracted people with minimal legal protection based on ageing, sexuality and/or gender diversity leading to isolation and feelings of insecurity by older lesbian women [30,39,44]. In addition some studies reported that ageing lesbian women believed that they did not fit within their communities [39] and experienced stigma and discrimination at work places, restaurants and night clubs [39,44]. Consequently lesbian women, and especially those of advancing age, became hesitant to ask for support when needed [35], hid their sexual identity when they were in public places [39] and preferred to interact only with other lesbian women within 10 years of their own age [31,39].

### 3.3. Institutional: Is Aged Care Inclusive?

The second theme identified from the syntheses focuses on assessing the inclusiveness of aged care in relation to their policies, advertising materials, physical configurations of their aged care spaces, provider knowledge and attitudes towards GSD individuals. Overall, a majority of studies revealed that aged care lacks inclusiveness with regard to the needs of GSD communities [1,41]. For example, a study showed that aged care assessment tools did not include LGBTI identities and needs, and brochures used to advertise aged care facilities either “omitted” or “silenced” sexual orientations and genders other than the dominant heterosexuality [1]. Whilst some brochures mentioned diversity referring to people from non-English speaking backgrounds, diversity concerning sexualities was not recognised [1]. Concerning aged care spaces, options were limited for all couples wishing to stay together regardless of their sexual orientation, particularly if they required different levels of care [1,41]. This was due to the assumption that older people, regardless of their sexuality, are “not sexual beings” [41]. However, the dominant heterosexual discourse and heteronormative attitude contributed for the exclusion of homosexual spaces and needs in the construction of aged care facilities [1,41]. The lack of inclusiveness of aged care impacted GSD women’s interaction with providers and their access to aged care. For instance, due to the lack of welcoming providers and aged care facilities, lesbian women lacked interest in revealing their sexuality to providers when accessing aged care [38]. In addition, they showed preference to lesbians’ [38] or women’s only aged care facilities [1] not to be marginalised and excluded. Other researchers reported the lack of interest by lesbian women to access aged care [38]. Finally, integration of aged care into the community or community ownership was recommended to make aged care facilities more welcoming, inclusive and non-discriminatory [41].

### 3.4. Societal Barriers to Care

#### 3.4.1. Heteronormative Assumptions

A majority of the reviewed papers reported that the presence of heteronormative assumptions in the health system compromised the type and content of discussions lesbian women would have with health care providers [32,33,38]. Some papers also reported frustrations of lesbian women having to go through the routine screening questions which made an assumption of heterosexuality [32] and that they were grateful for practices that included lesbian and other identities in their screening questions.

#### 3.4.2. Financial Barriers

Access to care was restricted for lesbian women living in rural areas, and the cost of transport was an issue when traveling to the cities to get care [30]. For women who needed additional home care hours, lack of financial resources was a barrier [35], as women were on social assistance and welfare would not cover it [46], so the service was not financially viable to them. In addition, Reference [31] reported that older lesbians experienced financial difficulties that included not having access to social security, pensions and health insurance, which significantly impacted their contact and utilisation of services they require in day-to-day life. Within the demographic of older lesbian women, the common instance where the individual had multiple chronic health conditions further compromised their access to health and social care [35].

#### 3.4.3. Health System Issues

The healthcare system focused on the biomedical model of health which focuses on disease rather than wellness [30,34,35]. Some papers reported health care providers’ lack knowledge about lesbian health, which was exemplified in the prescribing of contraceptive pills despite being aware of their GSD status [30,46].

## 4. Discussion

The review indicates that older GSD women experience challenges when accessing health and aged and social services at all levels of the socio-ecological arena. This suggests that interventions are needed to improve access and quality of care to older GSD women.

At the individual level, the intersections of age, gender and sexuality greatly impact older lesbian and bisexual women’s experiences with health, aged and social care. For instance, various age-related health conditions have been reported to compromise older lesbian and bisexual women’s accessibility of health and social care [35]. Additionally, there exists an assumption that older individuals are asexual, thus impacting the range of care they are provided [6,41]. Described as an invisible minority [5], older lesbian and bisexual women experience a significant impact from the intersections of their triple minority status—age, gender and sexuality [20]. Considering the implications at the individual level alone, their status as an invisible minority is reinforced, and it can be argued that their current care providers do not adequately meet the needs of ageing GSD women. Therefore, there exists a dire need for a re-examination of the quality of care provided at health, social, aged care and retirement services in order to address these needs.

Furthermore, examining the challenges experienced by older lesbian and bisexual women at the interpersonal level introduces homophobia, dilemma of disclosure and connection/belonging in relation to both seeking and receiving support. There exists a mix of experiences in relation to disclosure of sexual orientation to care providers, where some perceived disclosing their sexual orientation as integral while others chose not to [32,34,38]. Non-disclosure often resulted in consequences such as loss of happiness and self-esteem [39] and extraneous investigations [32]. Although disclosure often addressed these issues [33,34,38], some individuals received negative reactions such as prejudice, discrimination and termination of services from their care providers [32,33,34]. Additionally, the connections older lesbian and bisexual women have with their family and friends, other GSD individuals and the wider community impact the care they are likely to receive. For instance, strong positive connections with family and friends often resulted in care being provided by their family and friends [31,40,43], while negative connections often conceived homophobic attitudes and non-support [35]. Both actual and expected discrimination is argued to be one of three major barriers in relation to GSD individuals’ experiences accessing health care services and is one of many concerns facing GSD individuals [8]. For instance, more than 50% of health care providers have reported that they are uncomfortable dealing with GSD women’s issues [9]. Alongside the current paper’s findings, it can be maintained that older GSD women’s experiences accessing health, social, aged care and retirement services is greatly impacted upon by their interpersonal challenges. Although it may not be possible for all care providers to address their own attitudes toward non-heterosexual identities, it can, however, be argued that care providers who possess positive attitudes toward LGBTQI individuals can ameliorate lesbian and bisexual women’s expectations of homophobia and their dilemma of disclosure by making themselves transparent in regards to LGBTQI-friendly policies (e.g., displaying a rainbow flag). In Australia for example, there is a national Rainbow Tick accreditation program which recognises organisations committed to safe and inclusive service delivery to LGBTI people and such initiatives can meaningfully improve GSD women’s access and utilisation of services [47].

Another significant finding revealed that the challenges at the institutional level impacting lesbian and bisexual women’s experiences in accessing aged care and retirement services include the lack of inclusiveness of such institutions. A majority of aged care facilities ignore the issues of diversity regarding gender and sexuality [1,41]. Limitations in the inclusivity of such services included the aged care spaces available and care provider knowledge and attitudes, with a majority of studies reporting policies and advertising materials omitting non-heteronormative sexualities and genders [1]. Furthermore, Horner et al. [12] and Sinding and colleagues [46] maintain that few services accommodate the specific needs of lesbian and bisexual women. It can, therefore, be argued that current institutions do not adequately address the specific needs of older GSD women and that there is a dire need to re-work the policies and programs of such systems. By addressing these needs, there will be a much higher number of LGBTQI-inclusive services available for older GSD women.

Additionally, as indicated in this review, older lesbian and bisexual women currently accessing health, social and aged care remain hidden and are challenged by the dominant societal understandings of gender and sexuality. Due to homophobia [30,34,46], fear of discrimination [46], heteronormative assumption and dominant heterosexual discourses [32,33,38], older lesbian and bisexual women may not be comfortable disclosing their gender and/or sexual identity to providers. Consequently, older lesbian and bisexual women have been reported to enter a heteronormative life and are forced back into the closet [13]. This can have severe implications on their health and access to appropriate care. It is, therefore, argued that current policies and programs be addressed in order to provide LGBTQI-inclusive practices, thus improving the quality and care provided at health, social, aged care and retirement services—that is, offering LGBTQI-specific services and eliminating unnecessary procedures (e.g., pregnancy tests).

This review has strengths and limitations that should be noted. This review is pioneering in examining the health, social and aged care experiences of older GSD women to inform practice and policy and identifying additional areas of research. However, generalisation to all older GSD women cannot be made as the studies included in this review exclusively focused on lesbian and bisexual women. Further, not all of the included papers delineated between the results of older and younger participants making extraction of only older women’s experiences difficult. This presents an important confounding factor in exploring the intersectional experience of ageing gender and LGBTQI identity that is often lost within broader data sets. Additionally, only searching for articles in English meant that studies published in other languages might not be included in the review. It could also be possible that some studies may have been missed as key words were only searched in article titles thus limiting the potential breadth of the findings. The review included both peer reviewed articles as well as unpublished reports that contain information related to the health, social and aged care experiences of GSD women, which makes publication bias unlikely [48]. To minimise the role of subjectivity in the review process, the article selection, data extraction and analysis processes were closely monitored by the second author.

Overall, there is a lack of research among GSD women. Past research has reported a number of challenges to conducting research on the LBTQI community, including the difficulty of recruitment due to their smaller proportion and their reluctance to respond to questions related to gender and sexuality [25]. This review revealed that research to date on GSD women’s experience with health, aged and home care services has exclusively focused on lesbian women with only two studies including bisexual women. This means that the views and experiences of bisexual, transgender, queer and intersex women in accessing services remain largely unknown, which suggests that further research is needed. This is particularly important as the number of older bisexual, transgender, intersex and queer women increases as the general population ages.

## 5. Conclusions

The review indicates that older GSD women experience multiple and multilevel challenges when accessing health, aged and social services at all levels of the socio-ecological arena. This suggests that interventions are needed at all levels to improve access and quality of care to older GSD women.

## Figures and Tables

**Figure 1 ijerph-17-03861-f001:**
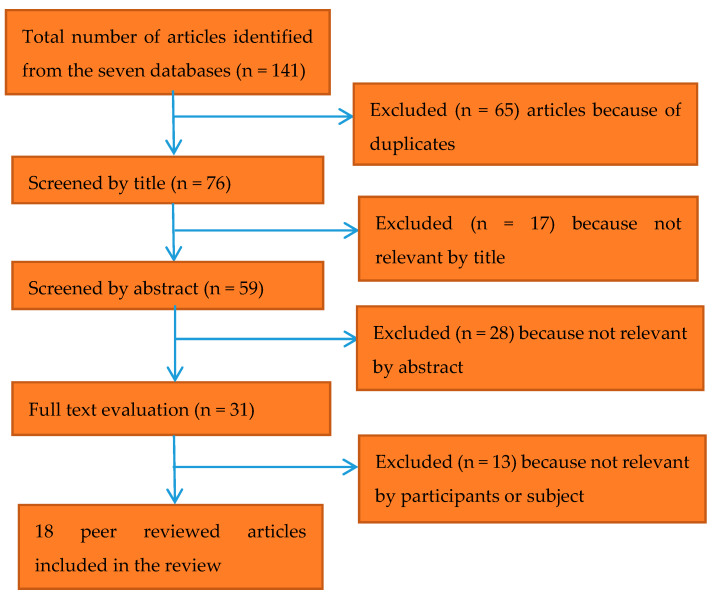
Article selection process.

**Figure 2 ijerph-17-03861-f002:**
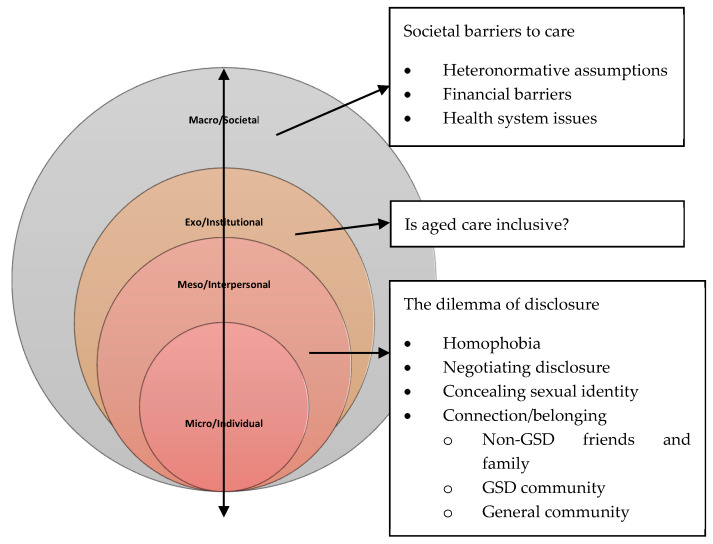
A socio-ecological analysis of the experiences of gender and sexually diverse (GSD) women accessing health, social and aged care services.

**Table 1 ijerph-17-03861-t001:** Summary of the inclusion/exclusion criteria and keywords.

Parameters	Inclusion	Exclusion	Key Words/Steps
Context	International	None	N/A
Language	Written in English	Other languages	Select for English only
Time	1990	Before 1989	Select from 1990 onwards
Population	Studies which include older lesbian, transgender, bisexual, intersex and queer population	Studies which only focus on young LGBTQI population	Older OR ageing OR aged OR elder * title OR aging (title) and Title (lesbian * OR transgender OR transsexual OR bisexual OR intersex OR queer OR homosexual * OR ‘same sex attracted’ OR ‘non-heterosexual’ OR ‘sexual minorities’ OR ‘gender diverse’ OR ‘sexually diverse’ OR two-spirit)
Interest	Studies concerned with the participants’ experiences (i.e., views and perceptions including facilitators and barriers) of accessing health, social and aged care services	Not concerned with health, social and aged care	(Title) AND ‘health need’ (title) OR ‘health access’ (title) OR ‘health care’ (title) OR ‘health services accessibility’ (title) OR ‘social service’ (title) OR ‘aged care ‘(title) OR retirement (title) ‘residential care’ (title) OR ‘nursing home’ (title) OR ‘home care’ (title) OR ‘community membership’ (title) OR belonging (title)
Study type	Primary research including qualitative, quantitative and mixed method designs	Studies which DO NOT include qualitative, quantitative and mixed methods of data collection and analysis	NA
	Grey literature (unpublished documents)	Book reviews, letters to the editor, editorials, opinion pieces, literature reviews, policy documents	NA

**Table 2 ijerph-17-03861-t002:** A comparison of demographic and methodological characteristics of individual studies included in the systematic review.

No	Authors/Year	Country	Participants	Sample Size	Age Range	Services Referred	Research Design	Data Collection	Theoretical Approach	Main Findings
1	Anderson et al., 2001 [30]	Canada	Lesbian	40	18+	Health care	Qualitative	Focus group discussions	-	Homophobic responses from health care providers and heterosexism discouraged lesbian women from accessing health care.
2	Averett et al., 2011 [31]	USA	Lesbian	456	51–86	Social and health care	Quantitative	Online survey	-	Older lesbians underutilised health and social services because of homophobia and ageism.
3	Barbara et al., 2001 [32]	USA	Lesbian	32	18 to 55+	Health care	Qualitative	Focus group discussions	-	Lesbian women have anxiety and concern related to self-disclosure of sexual orientation; non-disclosure of sexual orientation; seeking out gay-positive physicians; frustration with assumptions of heterosexuality; treatment of lesbian partners.
4	Bjorkman et al., 2009 [33]	Norway	Lesbian	121	18+	Health care	Qualitative	Written answers to a web-based open questionnaire	Heteronormativity	Health care professionals should facilitate the disclosure of a lesbian orientation; display a positive attitude towards homosexuality; and acknowledge and respect the lesbian orientation in providing care to gender and sexually diverse (GSD) women.
5	Grigorovich, 2015 [34]	Canada	Lesbian and bisexual	16	55–72	Home care	Qualitative	In-depth interviews	-	Reveling sexual identity to home care workers involved a complex decision-making process and was done on a case-by-case basis.
6	Grigorovich, 2015 [35]	Canada	Lesbian and bisexual	16	55–72	Home care	Qualitative	Semi-structured interviews	Feminist political economy framework	Chronic illness, limited functional status and homophobia influenced older lesbian women’s ability to access support and care.
7	Grigorovich, 2016 [36]	Canada	Lesbian and bisexual	16	55–72	Home care	Qualitative	Semi-structured interviews	Feminist ethic of care	Quality of care was enabled when providers were attentive and responsive to lesbian and bisexual women’s needs, demonstrated appropriate competencies and actively enabled recipients’ comfort.
8	Hash et al., 2009 [37]	USA	Lesbian	2	69 and 77	Social care	Qualitative	Case studies	Psychological contracting	Older lesbian women experience isolation which impacted their access to support.
9	Huges et al., 2015 [38]	Australia	Lesbian	4	59–72	Health and aged care	Qualitative	Case stories/Narrative research	Socio-linguistic	Older lesbian women have diverse perspectives about disclosing sexual identity, socialising with lesbian groups and accessing aged care.
10	Jacobson et al., 1998 [39]	USA	Lesbian	16	60+	Discrimination and leisure	Qualitative	Written responses to leisure questions and in-depth interviews	Ecological perspective	Discrimination and stigma influenced older lesbian’s leisure.
11	Rowan et al., 2014 [40]	USA	Lesbian	20	50+	Health care	Qualitative	Interviews	Phenomenology	Close connections with family members and involvement in lesbian oriented groups are vital to accessing support.
12	Phillips et al., 2007 [1]	Australia	Lesbian	6	45–69	Aged care	Qualitative	FGDs and advertisement brochures	Feminist poststructuralism	Provision of services for older lesbians is structured in such a way that lead to exclusion and aged care advertising materials exclude non-heterosexual relationships.
13	Phillips et al., 2006 [41]	Australia	Lesbian	6	45–69	Aged care	Qualitative	Focus group discussions	Feminist poststructuralism	Aged care spaces are constructed to serve a normative understanding of identities and relationships by not meeting the needs of ageing lesbians.
14	Quam, 1997 [42]	USA	Lesbian	2	81 and 77	Home care	Qualitative	Case studies	-	Older lesbian women socialise little and have no interest in attending activities for older GSD people.
15	Richard et al., 2006 [43]	USA	Lesbian	25	55+	Social services	Qualitative	In-depth interviews	-	Ageing lesbians have financial and housing concerns and are less likely to access social services due to perception of bias within the service and lack of connections with service users.
16	Rowan et al., 2015 [44]	USA	Lesbian	-	52+	Health care	Qualitative	In-depth interviews	Intersectionality	Older lesbians experience discrimination at work and public spaces because of their sexual identities.
17	Sinding et al., 2007 [45]	Canada	Lesbian	26	36–72	Cancer care	Qualitative	Semi-structured interviews	Grounded theory	Older lesbians experience isolation and disconnection and receive support mainly from lesbian partners and friends.
18	Sinding et al., 2004 [46]	USA	Lesbian	26	36–72	Cancer care	Qualitative	Interviews	-	Heterosexism and homophobia impacted lesbian women’s access to standard care.

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
