# Peer review of "Are Services Inclusive? A Review of the Experiences of Older GSD Women in Accessing Health, Social and Aged Care Services"

_ijerph, 2020, doi:10.3390/ijerph17113861_

Round 1

Reviewer 1 Report

The authors have taken the time to engage with my comments and I hope this has been a constructive process for them.

There are some comments where the authors have responded to my queries, but not yet incorporated these changes into their manuscript. For example, the authors explain "We used an inductive approach to coding and analysing the data and “first order codes” were identified using the language that was being coded." - this has yet to be included into the manuscript. 

Further, the authors responded to my query about including articles that included younger people by discussing the modification of their inclusion/exclusion criteria. My particular concerns were about how this was controlled for during "when coding/extracting data", specifically, rather than at the level of inclusion fundamentally. This is discussed as a limitation in the discussion, but it would be useful to understand practically and methodologically how the authors worked around this during their 'inductive approach to coding'. An additional sentence or two here would be helpful for others to understand the process.  

However, besides these two minor points, the article remains exceptionally well written and contributes to an important gap in current knowledge, and following these fairly minor additions to methodological detail, should be eminently publishable by IJERPH. 

Author Response

Point by point response to reviewer 1

  1. The authors have taken the time to engage with my comments and I hope this has been a constructive process for them. There are some comments where the authors have responded to my queries, but not yet incorporated these changes into their manuscript. For example, the authors explain "We used an inductive approach to coding and analysing the data and “first order codes” were identified using the language that was being coded." - this has yet to be included into the manuscript. 

Answer: We have now added the following sentence into the manuscript:

 An inductive approach to coding and analysing the data was used, and first-order codes were identified using the language that was being coded.

  1. Further, the authors responded to my query about including articles that included younger people by discussing the modification of their inclusion/exclusion criteria. My particular concerns were about how this was controlled for during "when coding/extracting data", specifically, rather than at the level of inclusion fundamentally. This is discussed as a limitation in the discussion, but it would be useful to understand practically and methodologically how the authors worked around this during their 'inductive approach to coding'. An additional sentence or two here would be helpful for others to understand the process.  

Answer: Only three of the 18 papers included in this review examined the experiences of a mixed group of younger and older lesbians. We had discussed to control the effect of “younger age” on the findings of the review during the data extraction, coding analysis and write up stages. However, it was not possible as the papers presented aggregated results/findings.

However, besides these two minor points, the article remains exceptionally well written and contributes to an important gap in current knowledge, and following these fairly minor additions to methodological detail, should be eminently publishable by IJERPH. 

Thank you for recommending our review for publications in the IJERPH.  

Reviewer 2 Report

The manuscript suffers from rigour related to replicability and the quality of a manuscript to be published in this journal.

Author Response

No specific comment is given by Reviewer 2. 

Reviewer 3 Report

This is a well-written review piece. I have several minor suggestions:

1) At the beginning of the manuscript, have you considered adding an introductory paragraph to highlight the significance of this review?

2) Among the 18 studies identified in this review, there are several studies that do not focus on older LGBTQ populations (as the authors define an older population as those aged 55 years and over, line 33 on page 1) e.g., Table 2, #1, #3, #4). I wonder if it is appropriate to include these studies in the review if the authors' focus is on the older LBTQI women.

Author Response

Point by point response to review 3

This is a well-written review piece. I have several minor suggestions:

  • At the beginning of the manuscript, have you considered adding an introductory paragraph to highlight the significance of this review?

Answer: we have now added a new paragraph describing the significance of the review (please see last paragraph of the background section):  

Findings of the review may contribute to engaging and informing decision makers to redesign health, social and aged care services to make it more accessible to older LBTIQ women and providing insights to foster appropriate care seeking behaviour.

  • Among the 18 studies identified in this review, there are several studies that do not focus on older LGBTQ populations (as the authors define an older population as those aged 55 years and over, line 33 on page 1) e.g., Table 2, #1, #3, #4). I wonder if it is appropriate to include these studies in the review if the authors' focus is on the older LBTQI women.

Answer: We have already justified the inclusion of papers that involved both older and younger LGBTQ women (please see last paragraph on page 3):  

Only a handful of studies initially addressed older gender and sexually diverse (GSD) (lesbian, transgender, transsexual, bisexual, intersex and queer) women’s views and experiences of accessing health, social and aged care services. As a result, the inclusion/exclusion criteria were modified to incorporate studies that included both older and younger gender and sexually diverse women.

This manuscript is a resubmission of an earlier submission. The following is a list of the peer review reports and author responses from that submission.

Round 1

Reviewer 1 Report

This is an exceptionally well written manuscript. The paper provides a solid rationale for it’s questioning, identifying an important gap in current knowledge, backed up by a good overview and use of literature. Review methodology is sensible, and well explained and documented. The authors account for the problems they experienced in their initial aims/scope and justify how these were sensibly overcome. Results are interesting, well presented, and important, though the focus on the three interlinked aspects of identity fades away slightly, with ‘age’ being slightly overshadowed.

I have some minor comments, suggestions, and queries:

The background section draws primarily on literature discussing the intersection of age and LGBTQI, the ‘third’ aspect discussed in this paper – gender – does not really appear until it is introduced as the third dimension of minority. It may be useful to include some literature that considers the intersection of age and gender more broadly.

The acronym GSD needs to be defined the first time it is utilised. In results, it seems to be abandoned and the term ‘older lesbian and bisexual women’ is used instead. Is this deliberate? If so, it should be explained, otherwise, consistency in terminology would be helpful to the reader.

Relatedly, at times, in the results, it may be worth reiterating that the study focusses on older GSD women. There is tendency to talk about ‘lesbian and bisexual women’ (177; 200; 240; etc), and ‘GSD women’ (190), ‘lesbian women’ (260; 275) without this additional qualifier. Given that the study’s focus is on the intersection of three aspects of identity, it may be worth being specific.

Indeed, in some sections of the results – for example ‘General Community’, the focus on ‘age’ completely fades away. Given the papers uniqueness being framed around the ‘triple minority’ – this needs to be brought back in and made explicit here. This could be done throughout the entirety of the results section really. What does the addition of the ‘third’ aspect of identity (age) change here – most of the discussions are around women’s sexuality, and its impact on accessing healthcare – what specific role does age play. The intersections between these three aspects need to be drawn out much more specifically – rather than age simply being implicitly in the background due to the sampled population. This is done much better in ‘Is Aged Care Inclusive’ where age plays an active part of the authors’ discussions, and indeed, in the discussion section itself, where the triplicate intersections are actively discussed. If this could be cascaded into the earlier results section, it would be much stronger and more consistent.

Regarding first order codes – I wondered where these arose from – where they generated from existing literature, based on the authors’ research questions, based on the researcher’s own expectations/interests/hypothesis, or using the language of what was being coded?

Regarding the inclusion/exclusion – how was ‘older’ defined (and controlled for) here. What counts as ‘old’?

Relatedly, with the inclusion of items like paper 1, which included the age range 18+, how did authors manoeuvre around this when coding/extracting data.

Line 234 – is this meant to be a new section? The formatting has gone awry here.

Likewise, Figure 2 is missing the caption of exo/institutional, and Figure 1 is also missing some text

Typo on line 263 ‘other y’

Line 235 describes a ‘second theme’, whilst line 281 describes a ‘fourth theme’ – where is the third theme?

The conclusion is very short. What are the implications of this article for future research agendas?

This may be my unfamiliarity with the journals style, but I wondered whether references could do with just a little tidying up/consistency. For example, I’m not sure why line 47-48 switches to in text citations instead of endnote citations.

Author Response

Thank you so much for reviewing our paper and providing feedback. We have revised the manuscript accordingly and uploaded a point-by-point response.  

Reviewer 2 Report

Thank you very much for allowing me to read the manuscript "Are services included? A systematic review of the experiences of older LBTQI women in accessing health, social and aged care services".
In general it is a pertinent manuscript but with a multitude of flaws. However, some suggestions are made to improve it.
Background: it is necessary either to justify or to reference because an individual with 55 years is considered to be older (line 33).
Line 47-49 correct what appears in parentheses
In lines 80-82, the phrase "Given increases in the ageing population (1) and average life expectancy (17), 80 coupled with identified concerns of ageing LGBTQI individuals (9), it can be argued that this review 81 provides a timely prompt to addressing an increasingly topical issue. Assessing its elimination
Include research design in the method.
In the search strategy it seems that the PICo strategy used is not adequate. Using the acronym PICO in this research to define the research question is not appropriate (https://www.ebsco.com/sites/g/files/nabnos191/files/acquiadam-assets/7-Steps-to-the-Perfect-PICO-Search-White-Paper.pdf ).
If there is a name that I do not know and that fits (it is possible) please indicate a bibliographic citation where it is explained and supported.
It is necessary to provide information on the search string of each of the databases as well as the results obtained in each of them.
To say that an exclusion criterion is research that is written in other languages methodologically is not correct and even less so in a systematic review that must be exhaustive. In fact it is a limitation that must be made explicit in this article.
Table 1 indicates as exclusion criterion "Studies which DO NOT include qualitative, quantitative and mixed methods of data collection and analysis", are there studies that are not included in that classification? I think it would be better to indicate "no letters to the editor, editorials, narrative reviews, etc.".
However, line 392 indicates that unpublished reports included in the review have been used. This has not been reported previously and changes the method.
In table 1 the search string for the interest parameter is incorrect.
The search strings only identify the key words in the title, when the logical thing is that they would have been searched in the title and abstract. It is recommended to redo the search or put as a limitation.
Lines 116 to 123 represent a conceptual framework that must be integrated in the "introduction" section justifying its relevance.

Who is the "first author"? (line 127) the author of a particular article or the author of the manuscript?
It is not correct in table 2 to indicate a bibliographic reference in the authors/year field. Add the information.
Table 2 should contain one more column with the synthesis of the most representative findings found in each research.
The results and discussion section should be shown (organized) from the point of view of the socio-ecological theory?
The result section is only descriptive and does not support the generation or strengthening of a theory. This is a limitation.
In line 388 it should be indicated that this revision is "pioneering" rather than "the first". Since it has not been searched in other languages one cannot be sure that it is the first one.
Limitations include that literature in languages other than English has not been reviewed with what this implies. This fact hides research on this subject.
In general, there are many formatting errors in the article that convey a very bad image. They give a bad feeling. They convey careless work. These formal aspects are very important. Figure 1 cannot be seen in its entirety. Authors' contributions are not included.
Quote 25 line 484 alignment defect
Bibliographical references do not conform to the format of the journal
Line 410 to 423 unformatted

Author Response

Thank you for reviewing our paper and providing feedback. We have revised the manuscript accordingly and uploaded a point-by-point response. 

Reviewer 3 Report

Manuscript Number: ijerph-674364

Full Title: Are services inclusive? A systematic review of the experiences of older LBTQI women in accessing health, social and aged care services

Thanks for having asked me to review this systematic review focused on examining the experiences of older LBTQI women in accessing health, social and aged care services

I have some comments that should be addressed to improve the manuscript.

Abstract

Page 1, line 15 What does GSD stand for? Getting Staff Done? Or Gender and Sexuality Diversity? Abbreviation in the abstract should be defined upon first use.

Page 1, line 18, is the manuscript a systematic review or a meta-synthesis? The authors included in the title that the manuscript is a systematic review, then in the abstract section, they used a different term: meta-synthesis. I suggest being coherent in using terminology because this could lead readers to misunderstand the manuscript.

Background

Page 2, line 52, what does STIs stand for? Sexually Transmitted Infections? Abbreviation in the text should be defined upon first use.

Methods

Did the authors conduct the meta-synthesis according to a protocol? Has the protocol been published or registered?

Page 2, line 85-87, the authors stated that articles were identified by searching eight databases. I can count just seven: 1) Scopus; 2) ProQuest Central; 3) Pubmed; 4) Psyinfo; 5) Informit; 6) ProQuest social science database; 7) SocIndex.

Page 2, line 86, Does “Psyinfo” stand for “PsycINFO”?

Page 2, line 88, PICO framework needs the reference.

Page 3, line 97, the authors should motivate why they limited their research to papers published after 1990.

Page 3, line 102, table 1 also presents inclusion and exclusion criteria as such I suggest specifying in the text such information.

Page 3, Table 1, the authors should specify terms and keywords that concern qualitative research interviews, focus groups and those that concern perspective and perception such as feeling, understanding, perspective.

Page 3, Table 1, Did the authors use filters of the databases for qualitative research?

Page 3, Did the authors also check each article’s reference list looking for new papers?

Page 3, It is not clear how the authors managed duplicates removal.

Page 3, According to which standards they reported the literature search? The 21-item Standards for Reporting Qualitative Research (SRQR)?
The 32-item checklist for interviews and focus groups titled Consolidated criteria for reporting qualitative research (COREQ)?

Page 4, line 112-114, According to the Cochrane Collaboration, a good meta-synthesis requires a critical appraisal of the papers included. One of the most frequently used instruments is the Critical Appraisal Skills Program qualitative study checklist, for example.

Results

Page 5, Figure 1, It is not clear which flow diagram the authors used for reporting of meta-synthesis: ENTREQ statement? The GRADE-Cerqual protocol? The Cochrane? EVIDENT?

Author Response

Thank you for reviewing our manuscript and providing feedback. We have revised the manuscript accordingly and uploaded a point-by-point response. 

Round 2

Reviewer 2 Report

The present version is improved and has partially taken into account some of the indications suggested above.

There are still methodological defects that make it doubtful whether the results are correct.

Author Response

Point by point response: 

  • Please add a completed PRISMA checklist (should be considered for publication as supplementary information)

Response: A completed PRISMA checklist has been attached in this round of revision.

  • Please add the Present full electronic search strategy for at least one database, including any limits used, such that it could be repeated. (should be considered for publication as supplementary information)

Response:  The search strategy we put in Table 1 (column 4 row 5) is the one we used for PubMed. Anyone interested should be able to use those key words and combinations, and repeat the search.

  • Correct the erroneous Figure 1 (numbers do not add up)

Response:  The number of articles excluded after screening by title was 17, not 21. The numbers now add up.

  • MAJOR POINT: Please make a major revision of how the results are presented. Currently, the Table 2 only reports methodological details and findings, but the Table does not give an overview of findings per study (as has been already requested by Reviewer 2 before).

Response: We have added a new column and summary of the main findings per study (please see table 2).

  • In the current version, the results text is long, narrative, poorly structured, unclear (number of themes do not add up), and the text is inconsistent with the Figure. The Figure should be used as an exact template for the subheadings in the results section, and then the Results text should be structured exactly according to the Figure.

Response: The Figure has now been revised to serve as a template of the headings and subheadings used in the results section.

  • It is not clear whether Figure 2 is novel, based on the findings of this search, and not a copy or a modified version of a Figure that has been published before elsewhere. This has to be clarified.

Response:  As we have stated in the Data synthesis and reporting section the socio-ecological framework was used to inform the analysis and interpretation of the review findings. We used the figure to demonstrate how the review finings fit into the model and the figure has not been published anywhere before.  

Reviewer 3 Report

The authors in the manuscript replaced the term "meta-synthesis" with "review"; they should modify the title, accordingly. The authors did not mention that the review was conducted according to a review protocol. The authors searched seven databases, but they did not modify the abstract accordingly. Page 3 line 120 the authors did not include in the manuscript why they limited their research to papers published after 1990. The authors did not mention that they checked each paper’s reference list looking for new papers. In the manuscript, the authors did not write that the Endnote software was used to remove duplicates in the manuscript. According to the authors’ decision not to perform critical appraisal of the papers included in the review, they can not say that their study is a systematic review. The manuscript has overall methodological weaknesses that do not make the results reliable and do not allow publication

Author Response

Point by point response to review 3

The authors in the manuscript replaced the term "meta-synthesis" with "review"; they should modify the title, accordingly. 

Answer: The title has now been modified to reflect the reviewer’s suggestion.  

The authors did not mention that the review was conducted according to a review protocol. 

Answer: As we have answered on our responses during the first-round revision, the protocol was a summary of the research question, key words including combinations and databases to be searched. It was drafted to guide the review process across all the authors.  

The authors searched seven databases, but they did not modify the abstract accordingly. 

Answer: The abstract has been modified to show that 7 databases were searched. 

Page 3 line 120 the authors did not include in the manuscript why they limited their research to papers published after 1990. 

Answer: We have now added the reason why we restricted the search period from 1990:

Initial search of articles revealed that literature on the health, social and aged care services experiences of older GSD women started to emerge around 1990. As such, the year of the search was agreed to start from this period.

The authors did not mention that they checked each paper’s reference list looking for new papers.

Answer: We have added the following sentence in to the manuscript to reflect this:

Finally, the reference list of all identified reports and articles were examined for additional literature.

In the manuscript, the authors did not write that the Endnote software was used to remove duplicates in the manuscript. 

Answer: We have already indicated in the manuscript that Endnote software was used to store and screen the articles. Please see page 3, line 113

According to the authors’ decision not to perform critical appraisal of the papers included in the review, they cannot say that their study is a systematic review. 

Answer:  We believe that we have done the search, screening, analysis and reporting of this review systematically and according to agreed standards. We have also justified the reason why we did not perform critical appraisal of the papers and cited a reference.